# The Early Oxidative Stress Induced by Mercury and Cadmium Is Modulated by Ethylene in *Medicago sativa* Seedlings

**DOI:** 10.3390/antiox12030551

**Published:** 2023-02-22

**Authors:** María Laura Flores-Cáceres, Cristina Ortega-Villasante, Pablo Carril, Juan Sobrino-Plata, Luis E. Hernández

**Affiliations:** Laboratory of Plant Physiology, Department of Biology, Universidad Autónoma de Madrid, 28049 Madrid, Spain

**Keywords:** alfalfa, cadmium, ethylene, heavy metals, mercury, oxidative stress

## Abstract

Cadmium (Cd) and mercury (Hg) are ubiquitous soil pollutants that promote the accumulation of reactive oxygen species, causing oxidative stress. Tolerance depends on signalling processes that activate different defence barriers, such as accumulation of small heat sock proteins (sHSPs), activation of antioxidant enzymes, and the synthesis of phytochelatins (PCs) from the fundamental antioxidant peptide glutathione (GSH), which is probably modulated by ethylene. We studied the early responses of alfalfa seedlings after short exposure (3, 6, and 24 h) to moderate to severe concentration of Cd and Hg (ranging from 3 to 30 μM), to characterize in detail several oxidative stress parameters and biothiol (i.e., GSH and PCs) accumulation, in combination with the ethylene signalling blocker 1-methylcyclopropene (1-MCP). Most changes occurred in roots of alfalfa, with strong induction of cellular oxidative stress, H_2_O_2_ generation, and a quick accumulation of sHSPs 17.6 and 17.7. Mercury caused the specific inhibition of glutathione reductase activity, while both metals led to the accumulation of PCs. These responses were attenuated in seedlings incubated with 1-MCP. Interestingly, 1-MCP also decreased the amount of PCs and homophytochelatins generated under metal stress, implying that the overall early response to metals was controlled at least partially by ethylene.

## 1. Introduction

The release of cadmium (Cd) and mercury (Hg) to the environment, caused mainly by mining, metallurgy, and other industrial activities, represents a major problem to the environment and human health [1,2]. Cadmium and Hg are extremely persistent pollutants and highly toxic to most living organisms, even at low concentrations [3]. Plants exposed to Cd and Hg suffer from growth inhibition, damages to the cellular metabolism, and, ultimately, cell death [4]. One of the major alterations caused by the accumulation of these metals is the induction of oxidative stress, through the generation of reactive oxygen species (ROS), such as superoxide (O_2_^●−^) and hydrogen peroxide (H_2_O_2_) [5,6]. This oxidative stress can be early induced by Cd and Hg, which is detected as rapid accumulation of H_2_O_2_ in roots after only 1 to 3 h of treatment [7,8]. Several cellular sources of ROS are envisaged, among them are the activation of plasma membrane NADPH oxidases that release apoplastic H_2_O_2_ [9,10], and mitochondria, where intracellular ROS are produced [11]. In turn, it is thought that apoplastic H_2_O_2_ generated under metal stress feeds extracellular basic POXs, which enzymatic activity augments under Cd and Hg stress [12,13]. These enzymes catalyse lignification and cross-linking of cell wall components, enhancing cell wall stiffness arresting cell elongation [14]. Additionally, the accumulation of ROS leads to drastic alterations in cellular components, causing lipid peroxidation and protein oxidation [15,16]. Oxidative damage of proteins under harmful environmental conditions affect fundamental metabolic processes, including accumulation of small heat shock proteins (sHSPs) [17]. sHSPs belong to a super-family of chaperones that are targeted to different cellular organelles and prevent aggregation of proteins under several types of abiotic stresses, induced by H_2_O_2_ [18,19]. It has become apparent that sHSPs are overexpressed in several plant species shortly after toxic metal exposure, possibly as a defence mechanism to limit the accompanying cellular damages [20,21,22,23].

Plants possess an efficient antioxidant machinery to maintain ROS cellular levels under tight control. The antioxidant system is comprised of enzymes such as superoxide dismutase (SOD), ascorbate peroxidase (APX), catalase (CAT), and glutathione reductase (GR), which mainly use reducing metabolites such as glutathione (GSH; γGlu-Cys-Gly), ascorbate (ASA) and NADPH [24]. Cadmium and Hg are thought to indirectly favour ROS production by modifying the activity of the mentioned antioxidant enzymes, or by altering the redox status of antioxidant metabolites, as occur with other metals [6]. However, these responses to metals are highly dependent on the plant species, organs studied, phenology status, and metal doses [25]. Oxidative stress symptoms generally appear in plants subjected to prolonged treatments with high metal doses when extensive cell damage is observed [26]. This could explain the remarkable inhibition of root APX activity in an alfalfa seedling treated with 30 μM Hg for 24 h, while APX activity augmented after 1 to 3 h of exposure [9]. In addition, metal specific stress signatures are detected in GR activity, which is induced by Cd but is strongly inhibited by Hg [27,28]. Therefore, it is fundamental to establish appropriate doses and intervals of treatment prior to the characterization in detail of the specific mechanisms of metal stress perception and tolerance [29].

Glutathione and homologous biothiols, such as homoglutathione (hGSH, γGlu-Cys-Ala) in legumes, play a central role in the cellular redox balance and metal stress tolerance [30,31]. The antioxidant capacity of cells depends on enough reduced GSH, which transits to oxidized glutathione (GSSG). Subsequently, GSH level is recovered at the expense of NADPH in a reaction catalysed by GR, resulting in a GSH/GSSG ratio of 10 to 1 under non-stressed growing conditions [32]. Under moderate metal stress, GR activity augments, perhaps reflecting shifts in the cellular redox balance [28,33], while changes in the GSH/GSSG are frequently detected only under acute toxic conditions and cellular poisoning at high metal concentrations [9]. On the other hand, GSH is a precursor of phytochelatins (PCs) and biothiols with the basic γ-(Glu-Cys)_n_Gly structure (usually *n* = 2–5), and is considered the main metal binding metabolite in several plant species [34]. Legume plants also accumulate homophytochelatins (hPCs; γ-(Glu-Cys)_n_Ala), which are synthetized from hGSH, and are able to chelate Hg and Cd [35]. The lack of PCs or hPCs synthesis by the depletion of GSH and hGSH cellular levels result in poor Cd and Hg tolerance [36,37], leading to severe oxidative stress [27,37,38].

Recent research efforts have been directed to understanding the primary cellular mechanisms mediating the perception of Hg and Cd in different plant species [39,40], and various genes involved in ethylene signalling and synthesis have been identified [41]. Similar responses were detected in *Arabidopsis* plants treated with Cd, which overexpressed 1-amicocyclopropane-1-carboxylic acid synthase (ACS) and the ETHYLENE RESPONSIVE FACTOR 1 (ERF1) and accumulated ethylene, suggesting generation of ethylene under metal stress [42,43]. Indeed, the synthesis of ethylene was detected in pea, mustard, soybean, and tomato plants exposed to different doses of Cd [22,44,45,46]. This response occurred in parallel with the induction of oxidative stress [45,47,48]. Thus, an interrelationship between ethylene, ROS, and GSH metabolism may be particularly relevant in tolerance to toxic metal stress [45,49,50].

Our aim was to compare the early antioxidant and stress-induced responses and the biothiol profile after Cd or Hg exposure, and to study the contribution of ethylene to these responses in the legume plant alfalfa using a *microscale* hydroponic system that allowed us to detect oxidative stress at the cellular level. Ethylene perception was inhibited by 1-methylcyclopropene (1-MCP) to define the specific mechanisms of oxidative stress triggered by Cd and Hg that are mediated by ethylene, which could help to optimize plant tolerance to these contaminants.

## 2. Materials and Methods

### 2.1. Plant Material and Hydroponic System

Seeds of *Medicago sativa* ‘Aragon’ were purchased from “Semillas Mur S.L.” (VAT B50773605), sterilized, and grown as described by Ortega-Villasante et al. [9] in Murashige-Skoog (MS) nutrient solution. After 24 h acclimation, seedlings were treated with CdCl_2_ or HgCl_2_ (3, 10, and 30 μM) for 3, 6, and 24 h. In the ethylene signalling inhibition experiments, alfalfa seedlings were grown in identical conditions, but were pre-incubated with 10 μΜ 1-methylecyclopropene (1-MCP) 24 h prior to metal exposure. 1-MCP was also added to the corresponding growing media during the selected Hg and Cd treatments. We used a microscale hydroponic system where 25 seedlings were grown per biological replicate of each sample, with at least four full independent experiments [7]. Shoots and roots were frozen in liquid nitrogen and stored at −80 °C until analysis (see Appendix A for further details).

### 2.2. Chemicals and Antibodies

All products were analytical grade. Ampholytes pH 3–10 (BioLyte 1631113) and protease inhibitor cocktail (P2714) were purchased from BioRad (Hercules, CA, USA) and Sigma (St. Louis, MO, USA), respectively. Rabbit polyclonal antibodies α-cAPX (AS06180), α-GR (AS06181), α-HSP70 cytosolic (AS08371), α-sHSP17.6 cytosolic Class I (AS07254), and α-sHSP17.7 cytosolic Class II (AS07255) were bought from Agrisera (Vännäs, Sweden). Goat α-rabbit IgG secondary antibody conjugated to HRP and Amersham Hyperfilm ECL kit were purchased from GE Healthcare Life Sciences (Little Chalfont, UK), and 1-methyl cyclopropene (1-MCP) was kindly provided by AgroFresh Inc. (Philadelphia, PA, USA).

### 2.3. Redox Enzymatic Activities in Gel

Enzymatic extracts were prepared with 0.5 g of frozen material by homogenisation in an ice-cooled mortar and pestle with 1 mL of extracting mixture (10 mL 30 mM MOPS at pH 7.5, 5 mM EDTA-Na_2_, 10 mM DTT, 10 mM ascorbic acid, 0.6% PVP, 10 μM PMSF and protease inhibitor cocktail). After 15 min centrifugation at 12,000× *g* at 4 °C, the supernatant was stored at −80 °C in single-use aliquots of 100 μL. Protein concentration was determined using the Bio-Rad Protein Assay, and protein loading was re-adjusted using denaturing polyacrylamide gel electrophoresis (SDS-PAGE; [27]) and Coomassie blue staining, prior to specific *in gel* enzymatic activity staining after non-denaturing polyacrylamide gel electrophoresis (ND-PAGE).

GR (EC 1.6.4.2) and APX (EC 1.11.1.11) activities were detected after ND-PAGE (10% polyacrylamide) were loaded, respectively, with 15 μg and 5 μg protein of shoot and root extracts, according to the procedures described by Sobrino-Plata et al. [12]. NADPH oxidase (EC 1.6.3.1) was detected after ND-PAGE (10% polyacrylamide) loaded with 10 μg protein, incubated in 250 mM Tris-HCl buffer (pH 7.5) supplemented with 0.5 mg/mL nitroblue tetrazolium chloride, 0.2 mM NADPH, 4 mM CaCl_2,_ and 0.2 mM MgCl_2_ [27]. POX isoforms were detected after isoelectric focusing (IF) ND-PAGE (6.5% acrylamide), using ampholytes that covered the 3.0–10.0 pH range, with buffers composition and electrophoresis conditions described by Ros-Barceló et al. [51]. Protein loading was 15 μg and 5 μg of shoot and root, respectively. POX activity was revealed by incubating the gels in a solution of 100 mM 4-methoxy-α-naphtol and 10 mM H_2_O_2_ in 50 mM Na-acetate buffer at pH 5.0 (see Appendix A for details).

### 2.4. Immunodetection by Western Blot

Western-blot immunodetection was performed after protein separation by SDS-PAGE [27]. Gels were incubated in transfer buffer (48 mM Tris-HCl, 39 mM glycine, 1.3 mM SDS, and 20% methanol pH 8.3) for 1 h, and transferred to a NT Biotrace nitrocellulose membrane (Pall Corporation, Port Washington, NY, USA) using the semidry transfer system from BioRad (Trans Blot SD Electrophoretic Transfer Semi Dry Cell), according to the manufacturer’s instructions. Membranes were blocked with 5% skimmed milk in TBS (200 mM Tris-HCl and 5 M NaCl, pH 7.5), incubated overnight at 4 °C with primary antibody (α-APX, 1:2000 dilution; α-GR, 1:5000; α-HSP70, 1:3000; α-sHSP17.6, 1:1000; α-sHSP17.7, 1:1000), and then incubated with the secondary α-rabbit IgG conjugated with horseradish peroxidase (α-IgG:HR; 1:20,000 in 5% skimmed milk in TBS). The chemiluminescent peroxidase reaction (Amersham ECL Plus™ kit) was recorded in a ChemiDoc™ XRS+ System (BioRad).

### 2.5. Analysis of Biothiols

Biothiols were analysed by HPLC in extracts prepared from 0.1 g frozen intact material in 300 µL 0.25 M HCl [7]. After 14,000× *g* centrifugation for 15 min at 4 °C, 100 µL of the clear supernatant were injected in a Mediterranea SEA18 column (5 μm, 250 × 4.6 mm; Teknokroma, San Cugat del Vallés, Spain), using an Agilent 1200 HPLC system (Santa Clara, CA, USA). Thiols were quantified after post-column derivatization with Ellman reagent, using N-acetyl cysteine (N-AcCys) as internal standard (250 μM final concentration) [12].

### 2.6. Root Extracellular H_2_O_2_ Generation

Extracellular H_2_O_2_ release was measured in root segments (1 cm) according to the method of Ortega-Villasante et al. [9]. After washing and equilibration in MS medium buffered with 2 mM MES at pH 6.0 in the dark for 1 h, root segments were individually placed in a 96-well microtitre plate containing the same medium supplemented with 50 µM Amplex Red (Molecular Probes, Eugene, OR, USA). Fluorescence was recorded at λ_exc_ = 542 nm and λ_em_ = 590 every 5 min for 6 h using a Synergy HT Biotek plate reader (Winooski, VT, USA).

### 2.7. Dye Loading and Confocal Laser Scanning Microscopy

Cellular oxidative stress degree was visualised by staining with 10 µM 2′,7′-dichlorofluorescein diacetate (H_2_DCFDA), and cell death was detected with 25 µM propidium iodide (PI), which were observed with Leica TCS SP2 confocal microscope (Wetzlar, Germany) as described by Ortega-Villasante et al. [7]. Microscopy images are representative observations of at least three independent experiments.

### 2.8. Cadmium and Mercury Concentration

Homogenised dried (100 mg) plant material (after 72 h at 40 °C) were digested in 1 mL of oxidative acidic mixture (HNO_3_:H_2_O_2_:H_2_O; 0.6:0.4:1, *v*:*v*) by autoclaving (Presoclave-75 Selecta, Barcelona, Spain) at 120 °C and 1.5 atm for 30 min, according to Ortega-Villasante et al. [7]. After dilution in miliQ water to 6 mL, Cd and Hg were quantified by ICP-MS NexION 300 (Perkin-Elmer Sciex, San Jose, CA, USA).

### 2.9. Image and Statistical Analysis

Densitometry analysis of the bands were performed using the ChemiDoc™ XRS+ System and ImageLab Software (BioRad), according to manufacturer’s specifications. Only relevant differences are presented, and representative gels of three independent assays are shown. A one-way ANOVA statistical analysis with post hoc Duncan test was performed using SPSS 17.0 (SPSS Inc., Chicago, IL, USA). Results were expressed as mean ± standard error, and differences were considered significant at *p* < 0.05.

## 3. Results

### 3.1. Early Stress Responses to Cd and Hg

Alfalfa seedlings exposed to Cd and Hg suffered remarkable inhibition of growth concomitantly with metal concentration and time of treatment (Appendix A). The strongest growth inhibition was observed in seedlings treated with Hg, after 3 h with 30 µM Hg, reaching the maximum inhibition (40%) after 24 h (Figure 1A). On the other hand, the growth inhibition was below 15% in seedlings treated with 30 µM Cd for 24 h (Figure 1B). Metal concentration increased in parallel to metal dose and time of exposure. However, Hg accumulation reached high values already after 3 h of treatment (50 to 75% of the concentration found after 24 h; Figure 1C), while Cd concentration augmented remarkably in seedlings exposed for 24 h, paralleling the Cd dose supplied in the medium (Figure 1D).

The amount of HSP70, sHSP17.7 Class II, and sHSP17.6 Class I was studied as biomarkers of cellular damage [21,23] induced by toxic elements by Western-blotting (Figure 2). The amount of HSP70 was not affected at any exposure time (Figure 2a,d,g), while a remarkable induction of sHSP 17.7 (Figure 2b,e,h) and sHSP17.6 (Figure 2c,f,i) was found under metal stress. The amount of both sHSPs increased sharply even after only 3 h of treatment, indicating that the accumulation of these types of stress-related chaperones is promoted early by metal stress. However, this increase was more remarkable for Hg-treated seedlings, independently of the isoform and dose supplied, showing saturation profile. Nonetheless, Cd induction was delayed and milder, especially for sHSP 17.6 (Figure 2h).

The greatest changes in redox enzymatic activities were found in the roots of alfalfa seedlings (Figure 3), rather than cotyledon seedlings (Appendix A). APX activity was severely inhibited in seedlings treated with 30 µM Hg after just 3 h, albeit APX protein levels did not change appreciably (Figure 3a,b). Additionally, an earlier and stronger inhibition of GR activity was detected after only 3 h of treatment with 3 µM (Figure 3c). GR inhibition was almost complete after 6 to 24 h of exposure to 10 and 30 µM Hg, with an extremely faint band observed in 3 µM Hg-treated seedlings (Figure 3g,k). Interestingly, the inhibition of GR activity by 30 µM Hg was accompanied with a higher accumulation of GR protein (Figure 3h,l). On the other hand, Cd treatments increased *in gel* APX activity only in short treatments (3 h) with the lowest concentrations (Figure 3b), while an increase in GR activity was only detected after 6 h exposure to 10 or 30 µM Cd (Figure 3g,k), without changes in the levels of GR protein (Figure 3h,l).

Cd exposure did not appreciably change alkaline POX activity (Figure 3m,o,q), but NADPH-oxidases were activated after 3 to 6 h with 10 and 30 µM Cd (Figure 3n,p). Interestingly, NADPH-oxidase activity also increased in seedlings given the lowest dose of Hg (3 µM) after 6 or 24 h treatment (Figure 3p,r). On the contrary, POX and NADPH-oxidase activities were strongly inhibited in seedlings grown with 10 and 30 µM Hg, even after only 3 h of treatment (Figure 3m–r). This strong phytotoxic effect was accompanied with clear changes in the protein band patterning in alfalfa seedlings treated with 30 µM Hg for 6 and 24 h, observed by Coomassie blue staining (Figure 3, lines L).

The exposure of alfalfa seedlings to Cd and Hg led to root extracellular H_2_O_2_ generation and induction of oxidative stress and cell death in root epidermal cells, in a timed and metal dose dependent manner (Figure 4). All metal treatments increased H_2_O_2_ generation after 3 h. Interestingly, the high production of H_2_O_2_ in roots of seedlings exposed to 10 and 30 µM Hg decreased in prolonged (6 and 24 h) treatments, even below the control seedlings (Figure 4A). On the other hand, the highest fluorescence intensity occurred in seedlings exposed to 30 and 10 µM Cd for 24 h, with intermediate extracellular H_2_O_2_ generation in those treated with 3 µM Hg for 24 h (Figure 4A). The acute phytotoxicity of Hg was confirmed in vivo by oxidative stress (H_2_DCFDA, pseudo-green colour) and cell death (IP, pseudo-red colour) fluorescent markers, in seedlings treated with 10 µM Cd or Hg for 6 h (Figure 4B). Scattered oxidative stress was detected in several root epidermal cells exposed to 10 µM Cd (white arrows), with limited numbers of dead cells (blue arrows). In the presence of 10 µM Hg, the number of necrotic cells augmented substantially, and oxidative stress (H_2_DCFDA staining) was only noticed in cellular layers below the epidermis (white arrow, lower panel Figure 4B). Control seedlings had few cells showing H_2_DCFDA staining, possibly due to mechanical stress in limited areas of the root epidermis (white arrow, upper panel Figure 4B).

The treatments of alfalfa seedlings with Hg and Cd also resulted in differential responses of the biothiols profile in roots, which depended on metal dose and time of exposure (Figure 5, Appendix A). However, minor changes were found in cotyledons, probably due to the limited exposure of the seedlings’ aerial part to Cd and Hg (Appendix A). Three biothiols were identified in alfalfa seedlings grown in control nutrient solution: cysteine (Cys), glutathione (GSH), and homoglutathione (hGSH). Homologous to GSH, these biothiols are present in many leguminous species [7]. The concentrations of Cys and hGSH were similar after 3 and 6 h, although after 24 h, hGSH became the major biothiol (over 45%, Figure 5 and Appendix A). Biothiol concentration decreased remarkably at the highest dose of Hg (30 µM), becoming virtually undetectable after 24 h (Figure 5), probably due to the strong binding of this metal with sulfhydryl group that impedes their detection with Ellman’s Reagent [35]. PCs were detected only after 24 h of metal treatments, specifically under moderate Hg doses (3 µM), with neat peaks of hPC_2_ ((γ-Glu-Cys)_2_-Ala) and hPC_3_ ((γ-Glu-Cys)_3_-Ala) (Appendix A). Under extreme Hg phytotoxicity attained with long (24 h) exposure, the amount of PCs decreased remarkably in 10 µM Hg treated alfalfa, and disappeared almost completely with 30 µM Hg (Figure 5, Appendix A). On the other hand, Cd promoted clearer changes in the biothiol profile at the highest dose (30 µM), in terms of PCs concentration (Figure 5, Appendix A) and variety, since PC_2_ ((γ-Glu-Cys)_2_-Gly)), hPC_2_, PC_3_ ((γ-Glu-Cys)_2_-Gly)), hPC_3_, PC_4_ ((γ-Glu-Cys)_4_-Gly)), and hPC_4_ ((γ-Glu-Cys)_4_-Ala)) were found (Appendix A).

### 3.2. Impact of Ethylene Signalling Inhibition on Metal Induced Stress

The influence of ethylene in the early responses of alfalfa to Cd and Hg was studied in seedlings pre-incubated with the ethylene signalling inhibitor 1-MCP (10 µM) for 24 h in the *microscale* hydroponic system. According to the results shown previously, Hg was more toxic than Cd, so we assessed ethylene regulation in metal doses that caused similar (moderate) phytotoxic effects: i.e., 3 µM HgCl_2_ and 30 µM CdCl_2_, after 6 and 24 h of treatment. Plants pre-incubated with 1-MCP suffered lower growth inhibition than their counterparts only treated with Hg and Cd (Figure 6).

Similar behaviour was found by the induction of several redox enzymatic activities upon exposure to Hg and Cd, which were remarkably attenuated when ethylene signalling was blocked (Figure 7). In particular, the specific inhibition of GR activity by Hg was reduced in 1-MCP pre-incubated seedlings, while the activation evoked by 30 µM Cd decreased when ethylene perception was blocked (Figure 7a). Additionally, 1-MCP prevented the activation of plasma membrane NADPH-oxidase in seedlings treated with 3 µM Hg and 30 µM Cd for 6 and 24 h (Figure 7e). However, APX activity did not change in response to metal stress or 1-MCP preincubation (Figure 7c), following the same pattern found in previous observations. Western-blot analysis using α-GR, α-APX and α-HSP70 did not show differences between treatments (Figure 7b,d,f), as observed previously. However, 3 µM Hg and 30 µM Cd caused strong induction of sHSP17.7 and sHSP17.6 after 6 and 24 h following the pattern already shown, which was nonetheless minimised in seedlings preincubated with 1-MCP (Figure 7g,h).

Extracellular H_2_O_2_ release increased concomitantly with time in seedling roots treated with 30 µM Cd and 3 µM Hg for 3, 6, and 24 h, with stronger response triggered by 30 µM Cd compared to 3 µM Hg (Figure 8A). Remarkably, AmplexRed fluorescence was attenuated in seedlings preincubated with 1-MCP for both metals, indicating that the production of H_2_O_2_ was limited when ethylene signalling was blocked. Similarly, intracellular oxidative stress augmented in Cd- and Hg-treated seedlings, with a scattered pattern of epidermal cells labelled by H_2_DFCDA. Interestingly, the oxidative stress and cell damage induced by both metals was again mitigated when seedlings were pre-incubated with 1-MCP and marked by the respective negligible signal of H_2_DFCDA and propidium iodide fluorescence in root epidermal cells (Figure 8B).

Regarding the total concentration of biothiols, they augmented in Cd-treated roots when exposure was prolonged from 6 to 24 h, whereas it remained close to control values under Hg after a transient depletion at 6 h (Figure 9A; Appendix A). As found previously, Hg and Cd led to a remarkable accumulation of PCs after 6 and 24 h of treatment, with hPC_2_ and hPC_3_ being the most abundant ones. The profile of biothiols detected was not affected by the inhibition of ethylene signalling, and only changes in their concentration were observed (Figure 9B). Pre-incubation with 1-MCP reduced the concentration of hGSH slightly under control conditions, while it augmented compared to single metal treatments when combined Cd or Hg plus 1-MCP were applied (Figure 9A, Appendix A). On the other hand, 1-MCP significantly diminished the amount of PCs that were produced under Cd and Hg stress: from 57% to 39%, and from 42% to 28% of total biothiol concentration, respectively (Figure 9A, Appendix A).

## 4. Discussion

Short-term exposure of alfalfa to Cd and Hg led to rapid root growth inhibition, accompanied by onset oxidative stress, as described previously [7,9,52]. Mercury was more toxic than Cd and triggered an oxidative burst at lower doses and time of treatment than Cd, in agreement with our previous results [9]. This inhibition could be related to alterations in the function and stability of biological membranes, in which different process alterations could modify the flow of H_2_O required for cell elongation [53]. For example, toxic metals impair ion homeostasis, leading to ion leakage and membrane depolarisation [54]. Under short-term treatments with Cd and Cu, plasma membrane H^+^-ATPase activity increased, possibly to restore the membrane potential required for proper transport across this membrane [55]. Such an alteration may lead to NADPH-oxidase activation and apoplastic H_2_O_2_ release in a process mediated by secondary messengers such as Ca^2+^ [56,57]. In this sense, we observed remarkable H_2_O_2_ production in roots of alfalfa seedlings upon short-term exposure to both Cd and Hg (Figure 3), as previously found in alfalfa and soybean seedlings [8,9]. Apoplastic ROS release is probably directed by plasma membrane NADPH-oxidases (Figure 3) which were activated under mild metal stress (Figure 2), following the pattern described by Montero-Palmero et al. [40]. However, under acute stress (i.e., 30 µM Hg) NADPH-oxidase (and peroxidase) activities were visibly inhibited, possibly reflecting extensive damages in membrane linked proteins as observed in roots of GSH-depleted genotypes of *Arabidopsis* sensitive to Cd- and Hg [37]. In addition, this is clearly visible in the Coomassie-stained protein band pattern under acute Hg stress (Figure 2L).

Different toxicity of Cd and Hg was also reflected in the enzymatic antioxidant and biothiol profile levels. Mercury caused stronger cellular damages than Cd in alfalfa seedlings, resulting in a remarkable inhibition of GR and APX activities in roots (Figure 2). The differential inhibition of GR activity, even at low doses of Hg, was firstly described in 15 day old alfalfa seedlings, possibly through an Hg-specific mechanism [28]. Interestingly, the strong inhibition of GR was accompanied by an accumulation of GR (Figure 2d,h,l), possibly as a compensatory mechanism. On the other hand, GR activity increase by Cd was also found in alfalfa [12] and wheat roots [33]. This differential response to Hg and Cd was also observed in the profile of GSH and PCs, with more diverse and enhanced accumulation of several PCs under Cd stress, in agreement with previous results in alfalfa and *Arabidopsis* [7,37].

Concomitant with oxidative stress and membrane integrity, we found a strong induction of sHSP17.7 and sHSP17.6 accumulation even at the lowest doses of Cd and Hg, while HSP70 amount did not change (Figure 1). Similar induction of those classes of sHSPs was found shortly after exposure of *Cucumis sativus* roots to Cd and Cu [58]. Conversely, heat shock pretreatments caused redox imbalance and activation of ROS scavenging enzymes, which, in turn, lessened Cd toxicity in rice seedlings, possibly through a H_2_O_2_ priming effect [20]. Cytosolic sHSP 17.2 was also induced by Cd in pea leaves, which was partially dependent on H_2_O_2_ [22]. H_2_O_2_ seems to activate several redox sensitive Heat Shock Factors (HSFs), transcription factors that promote the expression of sHSPs and ROS scavenging enzymes genes in several abiotic stresses, where NADPH-oxidases plays a relevant promotion role [59]. Interestingly, Shim et al. [60] found that HSF Class A4 is important for Cd tolerance, as it directs the expression of detoxifying genes such as metallothioneins. In recent years, it has been becoming clear the existence of a multi-level interplay between HSFs and ROS accumulation, with complex connections between protein chaperonins and ROS scavengers, two faces of the defence barriers triggered to limit cellular damage under oxidative stress [61]. Interestingly, these defence responses are apparently modulated by phytohormones such as salicylic acid or ethylene [62].

To test the hypothesis that ethylene may be involved in the early responses of alfalfa plants to Cd and Hg, we incubated a set of seedlings with the ethylene signalling blocker 1-MCP, known to delay senescence and oxidative stress symptoms under abiotic stress [63]. Although 1-MCP blocked the expression of ethylene responding genes and attenuated H_2_O_2_ release by alfalfa roots under metal stress [40], a comprehensive analysis of the antioxidant and tolerance responses mediated by ethylene was required in alfalfa seedlings. Thus, the early cellular oxidative stress, H_2_O_2_ apoplastic release, and NADPH-oxidase activity induced by Cd and Hg were remarkably attenuated by 1-MCP (Figure 5 and Figure 6), similar to previous results [40]. Likewise, the addition of aminoethoxyvinylglycine (AVG), an ethylene synthesis blocker, lessened the cell death rate in tomato suspension cells exposed to Cd [48]. Similarly, tomato plants with limited ethylene perception suffered lower oxidative stress symptoms caused by Cd [46]. Induction of oxidative stress arsenite (As(III)) toxicity was also overcome in ethylene signalling *Arabidopsis* plants, while the addition of the ethylene precursor ACC enhanced lipid peroxidation and As(III) damages [64]. Concerning Hg, GR activity may be considered to be a specific bioindicator of Hg toxicity in roots, as it declines severely in the presence of Hg (Figure 2). This coincides with previous results of Hg-sensitive *Arabidopsis* genotypes (*cad2-1* or *pad2-1* mutants defective in GSH synthesis), which showed stronger GR inhibition [27]. Therefore, the alleviation of Hg-dependent GR inhibition by 1-MCP may be due to attenuated stress symptoms occurring when ethylene signalling is blocked. Contribution of ethylene in the early oxidative stress responses to moderate doses of Cd, such as the overexpression of NADPH-oxidases, GSH synthesis and GR genes, were also described by Schellingen et al. [65] using *Arabidopsis acs2-1*/*acs6-1* double mutant defective in ethylene synthesis, and in *Arabidopsis etr1-1* and *ein2-1* ethylene signalling mutants [42]. Oxidative stress symptoms caused by Cd were also attenuated in the aquatic macrophyte *Nelumbo nucifera* G., and treated with the ethylene signalling blocker silver thiosulfate [66]. NADPH-oxidases contribute at least partially to the generation of ROS under Cd stress [67], which may explain the increased NADPH-oxidase activity detected in alfalfa seedlings exposed to Cd and Hg (Figure 2; [40]). Ethylene seems to facilitate the upregulation of NADPH-oxidase genes in the presence of Cd, which, in turn, modulates the expression of ALTERNATIVE OXIDASE1a, as observed in *Arabidopsis acs2-1*/*acs6-1* mutant and ethylene insensitive mutant *ein2-1*. This enzyme may contribute to generating ROS at the mitochondria, perhaps amplifying the oxidative stress signalling [68]. In agreement with these results, defective ethylene signalling also prevented the generation of H_2_O_2_ and O_2_^●−^, which are products of NADPH-oxidases caused by the exposure of Zn nanoparticles in leaves of *ein2-1* and *etr1-3 Arabidopsis* plants [69]. Finally, our results show that the attenuation of the oxidative stress obtained by 1-MCP under Cd and Hg stress also concurs with the limited expression of oxidative stress hallmark genes in *ein2-1* mutants in the presence of Cd [65].

Recent evidence supports the idea that H_2_O_2_ is an important component of metal stress signalling and important for the activation of detoxification mechanisms [70]. ROS prompt a series of redox-derived changes in protein function, leading to the quick alteration of gene transcription and activation of several signalling pathways that involve phosphorylation, Ca^2+^ release, and/or Cys residues redox shifts [71]. Among other processes, the overexpression of sHSPs depends on HSFs activation through a H_2_O_2_-mediated signalling cascade [72]. Conversely, the defence mechanisms triggered by HSF comprise the enhancement of ROS-scavenging systems, such as APX, which would contribute to maintaining the cellular redox balance under stress [73]. Additionally, ethylene prevents heat shock cellular damages, possibly through the activation of the heat shock pathway [74]. Ethylene seems to mediate in the basal thermotolerance of *Arabidopsis*, perhaps intertwined with ABA and other stress phytohormones, along with several antioxidants such as ASA and GSH, to prevent the oxidative stress caused by heat [75]. In this sense, the blocking of ethylene signalling with 1-MCP attenuated premature leaf senescence, ROS production and oxidative stress damages observed in soybean seedlings grown under high temperature stress [63]. This coincides with our data, as we found that 1-MCP decreased the early induction of ROS accumulation, oxidative stress, and enhanced accumulation of a sHSPs under Cd and Hg stress (Figure 5 and Figure 6). Therefore, our working model considers that ethylene enhances ROS production, and the subsequent redox unbalance may induce sHSPs expression under Cd and Hg stress.

Biothiols are keystones of plants’ tolerance to toxic elements [30], and crosstalk between GSH and ethylene is highly probable, since this phytohormone is synthesised from methionine and depends highly on S metabolism [49], and ethylene enhances sulphur assimilation [76]. For example, *Arabidopsis* mutant plants strongly defective in GSH synthesis (i.e., *rml1-1*) overexpressed ERF (Ethylene Response Factor) transcription factors ERF11, ERF2, and ESE3 [77]. On the other hand, ethylene, and other stress related phytohormones such as jasmonate and salicylic acid, seem to promote the biosynthesis of GSH or modify the GSH redox status [78]. Interestingly, we observed that incubation of alfalfa seedlings with the ethylene signalling blocker 1-MCP decreased the levels of GSH and PCs treated with Cd and Hg (Figure 7, Appendix A). In this respect, the activation of GSH synthesis promoted by Cd was blocked in mustard plants incubated with the ethylene synthesis inhibitor AVG [45]. Similarly, AVG repressed the early accumulation of GSH induced by Cd in *Lycium chinense*, which also affected the expression of GSH synthesis genes, such as γ-glutamylcysteine synthetase (γ-ECS) and glutathione synthetase [50]. Schellingen et al. [65] found lower expression of GSH metabolism genes in the ethylene defective *Arabidopsis* mutants *acs2-1*/*acs6-1* subjected to 24 and 72 h treatments with Cd. Additionally, the concentration of GSH in leaves also decreased in *acs2-1*/*acs6-1* mutant plants compared to the WT after 24 h, though the difference was less pronounced when Cd treatments were extended to 72 h. Similar results were observed in the ethylene signalling of *etr1-1*, *ein2-1,* and *ein3-1 Arabidopsis* mutants, where GSH levels were lower than the WT under Cd stress, which corresponded with the limited expression of glutathione synthase gene in the *etr1-1* genotype [42]. To sum up, ethylene seems to transiently mediate the canonical biothiol synthesis pathway under toxic metal stress. Although the signalling mechanisms are still unknown, we hypothesise that ethylene exerts its control on biothiols synthesis through an indirect pathway via a redox switch. Several GSH metabolic enzymes (i.e., γ-ECS and GR) required for GSH biosynthesis and homeostasis are post-translationally activated via Cys thiol/disulfide redox shifts under oxidative stress and redox cellular imbalance [79], which are apparently promoted by ethylene in the presence of toxic elements such as Cd and Hg.

It is becoming apparent that ethylene mediates the early activation by Hg and Cd of NADPH-oxidase and H_2_O_2_ production, as well as the differential changes in GR activity, i.e., inhibition by Hg and modest activation by Cd, and the synthesis of GSH and PCs; these are symptoms that were transiently delayed in alfalfa seedlings incubated with 1-MCP and exposed for 6 h to Hg and Cd. However, the exact mechanism by which ethylene promotes the induction of NADPH-oxidase activity and the alteration of the redox balance remains unknown, which could involve transcriptional regulation as well as post-translational activation, for example via calcium and calcium dependent kinases [80]. Future experiments should focus on analysing this aspect of the signalling process that occurs in the early moments of toxic metal exposure by tuning the response to ethylene supplied at different doses and times of incubation, and in the presence of 1-MCP to exploring the behaviour of plants at advanced phenological status (i.e., by comparing juvenile versus adult developmental phases). In addition, the attenuation of the metal-induced oxidative burst was less apparent after 24 h treatment, as Keunen et al. [68] observed that *Arabidopsis* plants lacking ethylene response (*ein2-1* and *ein2-5* mutants) suffered inexorably stress at higher Cd doses or longer treatments. These results imply that other mechanisms of toxicity perception, perhaps depending on other stress related phytohormones, such as jasmonate or abscisic acid [41], may overtake the ethylene dependent pathway as plants become poisoned by toxic elements, and this should also be a matter of future research. With the idea of improving metal tolerance, an interesting alternative to modify the endogenous levels of ethylene could be the inoculation of plants with bacteria producing ACC deaminase found to promote plant growth under Cd stress [81]. Therefore, such strategy could be combined with the selection of plants with limited sensitivity to and/or synthesis of ethylene.

## 5. Conclusions

Inhibition of ethylene signalling using 1-MCP resulted in attenuated induction of the oxidative stress triggered by Hg and Cd in alfalfa seedlings. Subsequently, ethylene seemed to modulate in the roots of juvenile alfalfa seedlings the generation of ROS, the accumulation of sHSPs (HSP17.7 and HSP17.6) chaperones, the induction of the pro-oxidant NADPH-oxidase activity, alteration of redox enzymatic activities needed to maintain the GSH redox balance, such as GR, and the ability to accumulate PCs under Cd and Hg stress. Characterisation of the role ethylene, and other stress-related phytohormones, will help to improving the tolerance of plants to toxic metals, with the aim of optimising phytoremediation strategies to clean up metal contaminated soils.

## Figures and Tables

**Figure 1 antioxidants-12-00551-f001:**
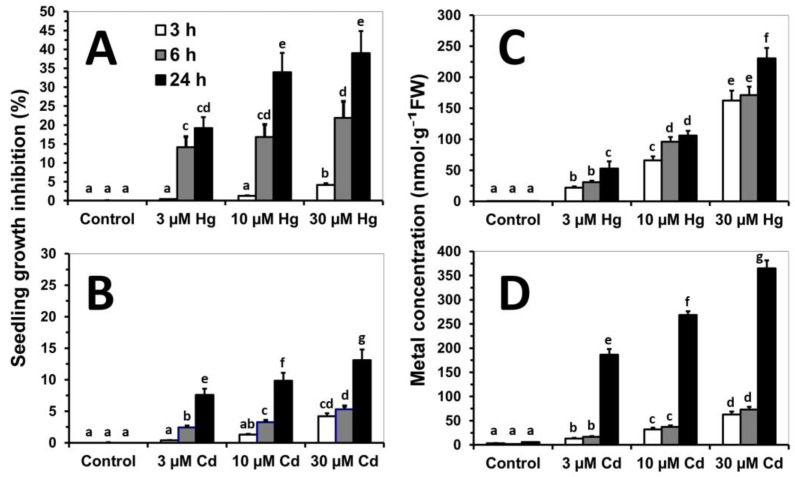
Relative growth inhibition (%) (**A**,**B**) and metal concentration (nmol g^−1^ FW) (**C**,**D**) of *Medicago sativa* seedlings treated with Hg (HgCl_2_) or Cd (CdCl_2_) (0, 3, 10, and 30 µM) for 3, 6, and 24 h in the *microscale* hydroponic system. The data are the average of four independent experiments, and the standard error (SE) bars are shown. Significant differences between treatments are shown with different letters (*p* < 0.05, Tukey’s test).

**Figure 2 antioxidants-12-00551-f002:**
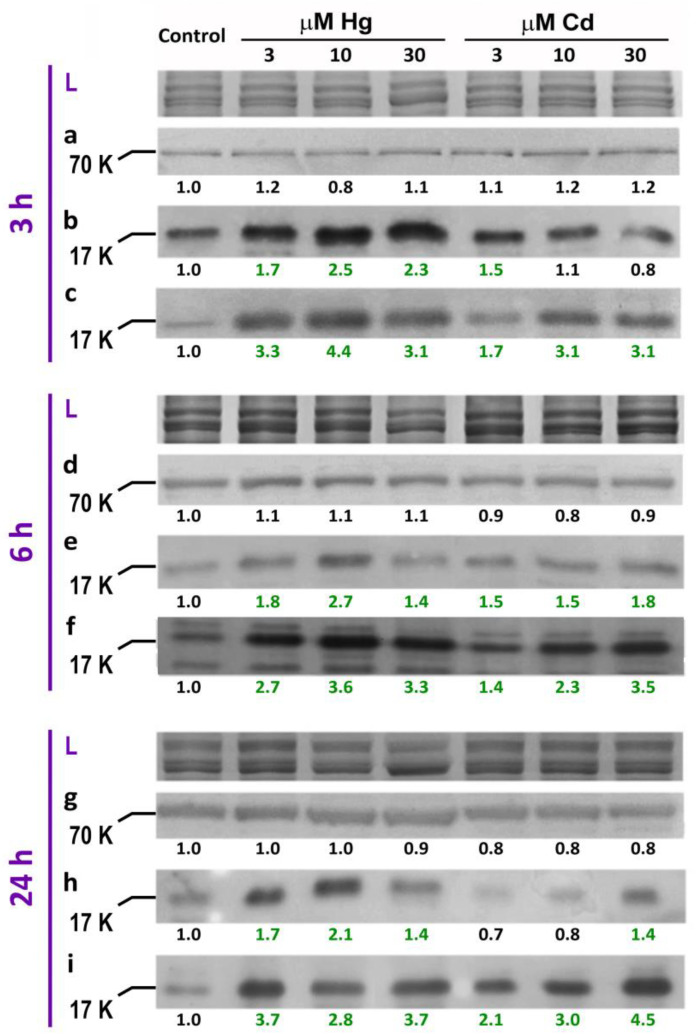
Induction of small Heat Shock Proteins (sHSPs) in roots of *Medicago sativa* seedlings treated with Hg or Cd (0, 3, 10, and 30 µM) for 3, 6 and 24 h in the *microscale* hydroponic system, detected by immunostaining: α-HSP70 (**a**,**d**,**g**), α-sHSP17.7 Class II (**b**,**e**,**h**), and α-sHSP17.6 Class I (**c**,**f**,**i**), with apparent molecular (K; KDa) weight of bands of interest. Band intensity was normalised against Coomassie blue general protein staining after denaturing PAGE, corresponding to L (protein loading) bands. The numbers show the fold change relative to control samples, and green coloured figures represent fold-changes above 30%.

**Figure 3 antioxidants-12-00551-f003:**
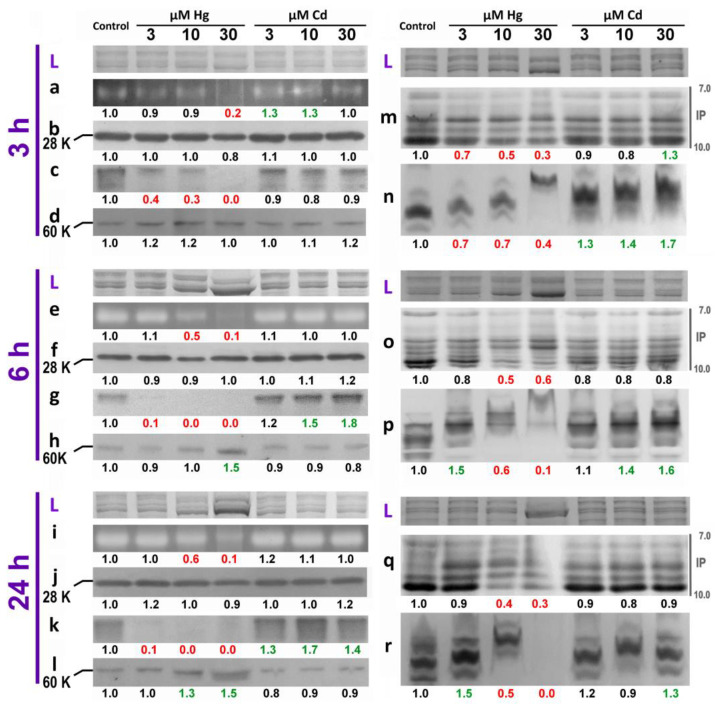
Redox enzymatic activities determined *in gel* after ND-PAGE, of alfalfa seedlings roots treated with 0, 3, 10, and 30 µM Hg or Cd for 3, 6, and 24 h. Identification of each band labelled with lower case letters: APX activity (**a**,**e**,**i**), α-APX (28 KDa apparent molecular weight) immunodetection (**b**,**f**,**j**), GR activity (**c**,**g**,**k**), α-GR (60 KDa apparent molecular weight) immunodetection (**d,h**,**l**), alkaline POX after isoelectric focusing at the immobilised pH (IP) 7.0 to 10.0 range (**m**,**o**,**q**), and plasmalemma NADPH-oxidase (**n**,**p**,**r**). Band intensity was normalised against Coomassie blue general protein staining after denaturing PAGE, corresponding to L (protein loading) bands. The numbers represent the fold change relative to control samples. Green (up) and red (down) represent fold-changes greater than 30%.

**Figure 4 antioxidants-12-00551-f004:**
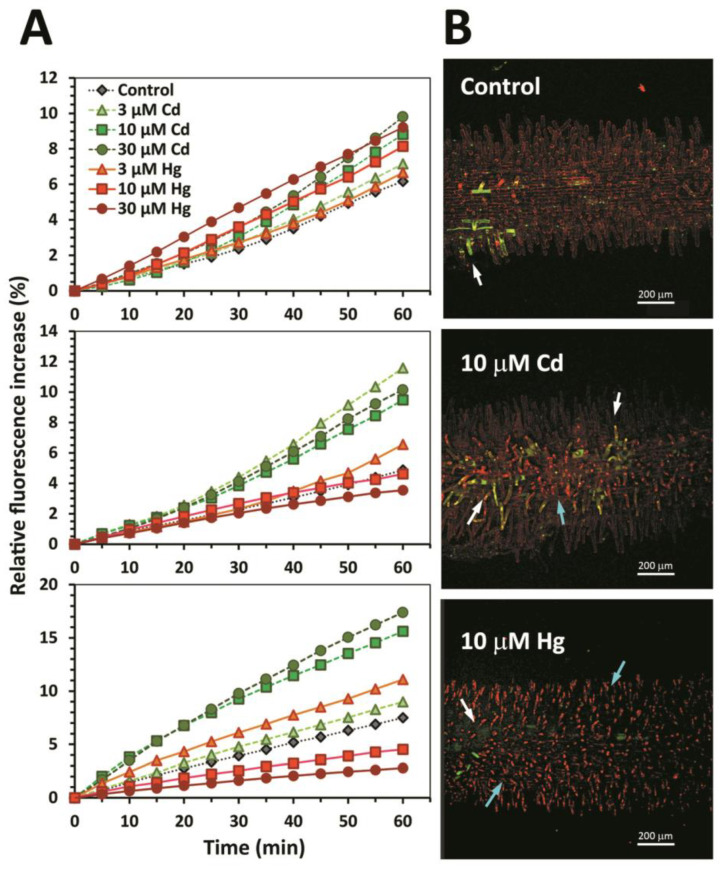
Generation of apoplastic H_2_O_2_ in alfalfa root segments and induction of oxidative stress in alfalfa root epidermal cells. (**A**) Measurement of AmplexRed fluorescence (%, relative to time 0) of root segments (*n* = 12 per treatment) collected from alfalfa seedlings treated with 0, 3, 10, and 30 µM Hg or Cd for 3, 6, and 24 h. (**B**) Confocal fluorescence microscopy to detect oxidative stress (H_2_DCFDA, green pseudo-colour, white arrows) in alfalfa seedlings treated with control, 10 µM Cd or 10 µM Hg for 6 h. Seedlings were counterstained with propidium iodide (PI, red pseudo-colour) to highlight cell walls and necrotic cells with condensed nuclei (blue arrows). Representative images of at least three independent experiments are shown.

**Figure 5 antioxidants-12-00551-f005:**
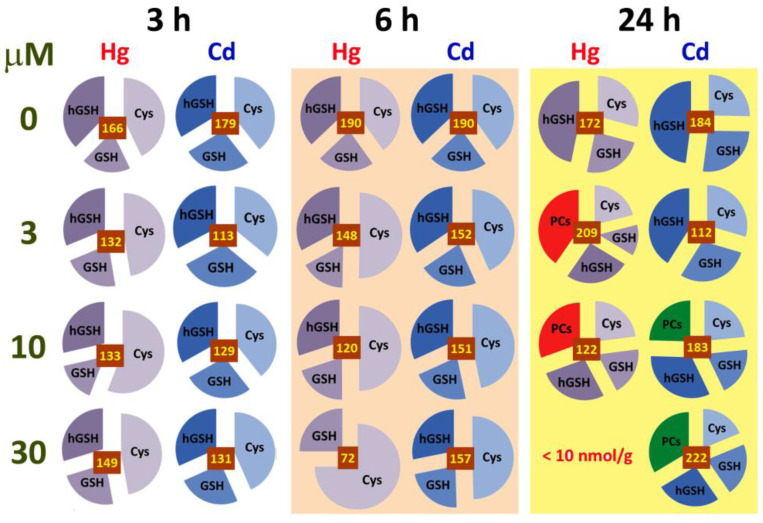
Biothiols profile in roots of alfalfa seedlings treated with 0, 3, 10, and 30 µM Hg or Cd for 3, 6, and 24 h. The relative abundance of each biothiol type is represented: Cys, cysteine; GSH, glutathione; hGSH, homoglutathione; and PCs, phytochelatins. The numbers in the brown box represent the average total concentration of biothiols (nmol g^−1^ FW) in each treatment. Absolute values and statistics are shown in Appendix A.

**Figure 6 antioxidants-12-00551-f006:**
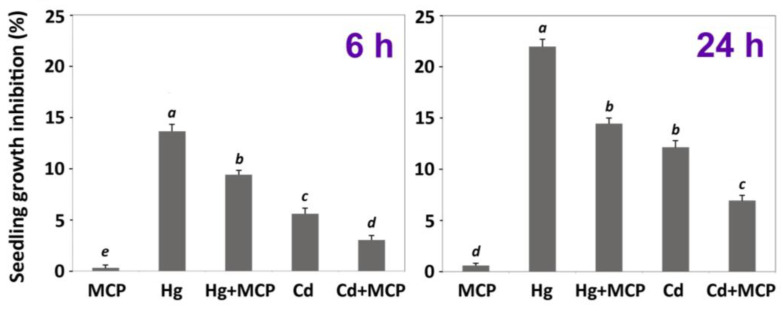
Functional analysis of ethylene-mediated responses to Hg and Cd. Growth inhibition of *M. sativa* seedlings treated with 3 µM Hg or 30 µM Cd and supplemented with 10 µM 1-MCP, after 6 and 24 h treatment. Significant differences between treatments are shown with different letters (*p <* 0.05, Tukey’s test).

**Figure 7 antioxidants-12-00551-f007:**
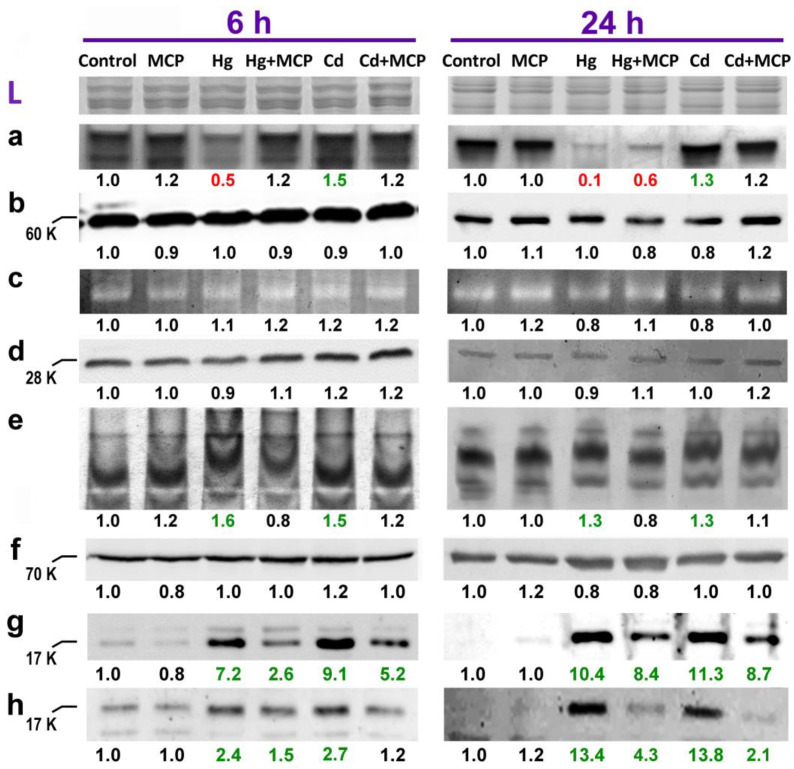
Functional analysis of ethylene-mediated responses to Hg and Cd in roots of alfalfa seedlings treated with 3 µM Hg or 30 µM Cd and supplemented with 10 µM 1-MCP, after 6 and 24 h treatment. Redox enzymatic activities and Small Heat Shock Proteins (sHSPs) immunostaining. Identification of bands labelled with lower case letters: GR activity (**a**), α-GR (**b**), APX activity (**c**), α-APX (**d**), NADPH-oxidase activity (**e**), α-HSP70 (**f**), α-sHSP17.7 (**g**), and α-sHSP17.6 (**h**). Apparent molecular weight (K, KDa) of proteins of interest is included on the left of the immunoblots. See legend Figure 3 for details of numbers’ annotations.

**Figure 8 antioxidants-12-00551-f008:**
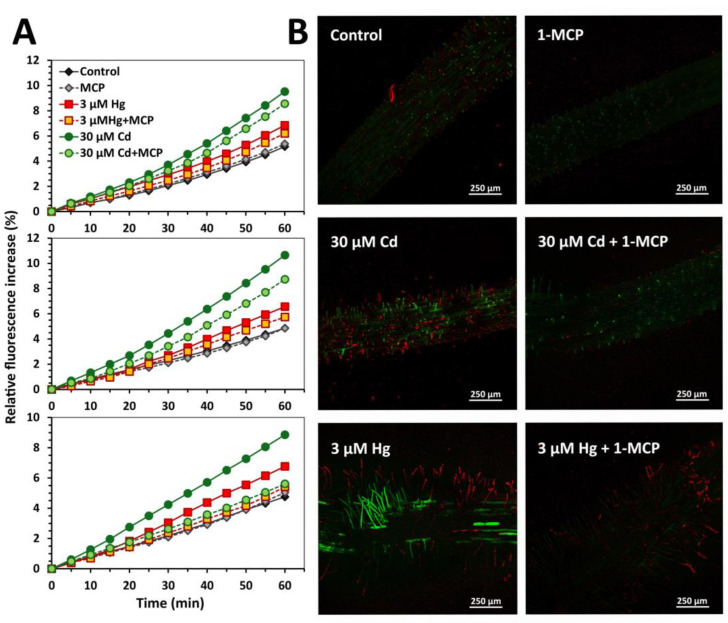
Apoplastic H_2_O_2_ production and intracellular oxidative stress induced by Hg and Cd are alleviated when ethylene signalling is blocked by 10 µM 1-MCP in *M. sativa* root segments. (**A**) Extracellular H_2_O_2_ production measured as Amplex Red fluorescence (%, relative to time 0) in alfalfa roots segments exposed to control, 3 µM Hg or 30 µM Cd in presence or absence of 10 µM 1-MCP, measured during 60 min. (**B**) Confocal fluorescence microscopy to detect oxidative stress (H_2_DCFDA, green pseudo-colour) in alfalfa seedlings treated as in (**A**) for 6 h. Seedlings were counterstained with propidium iodide (PI, red pseudo-colour) to highlight cell walls and dead cells. Representative images of at least three independent experiments are shown.

**Figure 9 antioxidants-12-00551-f009:**
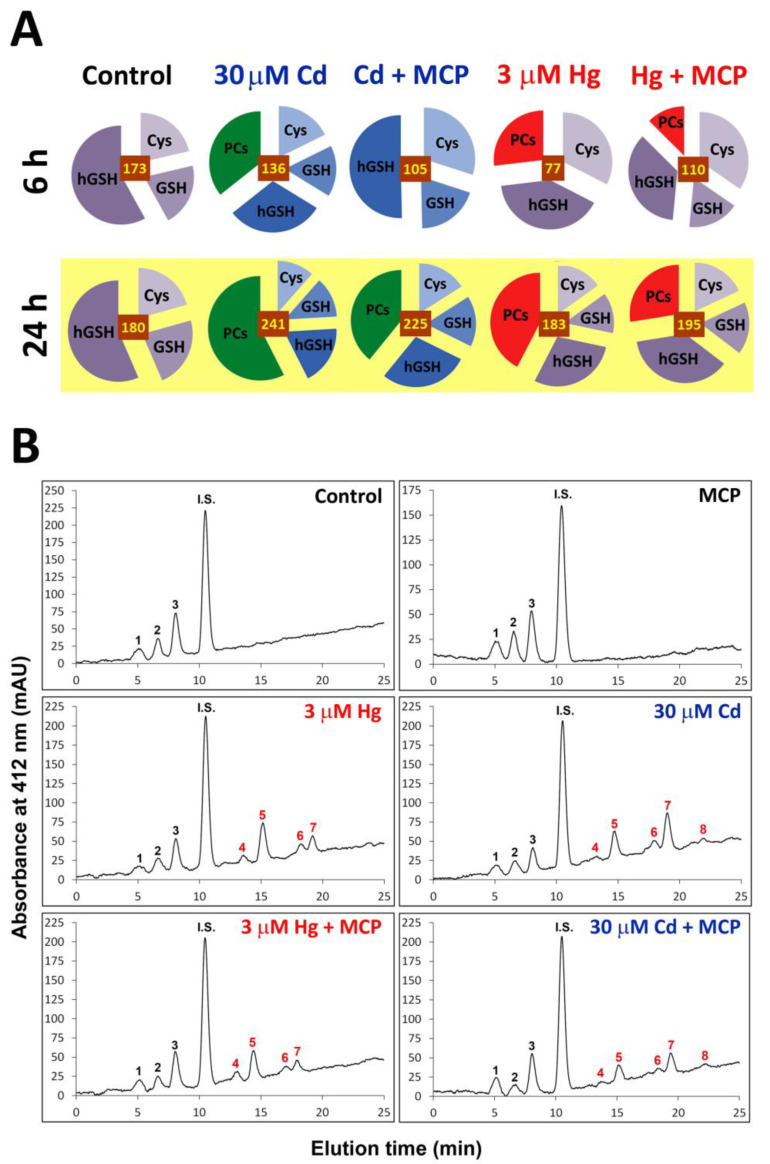
Influence of ethylene signalling inhibition (10 µM 1-MCP) on the biothiols profile in roots of alfalfa seedlings treated with 0 and 3 µM Hg or 30 µM Cd for 6 and 24 h. (**A**) The relative abundance of each biothiol type is represented: Cys, cysteine; GSH, glutathione; hGSH, homoglutathione; and PCs, phytochelatins. The numbers in the brown boxes represent the average total concentration of biothiols (nmol/g FW) in each treatment. Absolute values and statistics are shown in Appendix A. (**B**) Characteristic chromatographs of biothiols obtained in roots of alfalfa treated for 24 h. Black coloured peaks appeared in all samples: 1, Cys; 2, GSH; and 3, hGSH. In red, peaks detected in seedlings exposed to Hg or Cd: 4, PC_2_ (γ-GluCys)_2_-Gly); 5, hPC_2_ (γ-GluCys)_2_-Ala); 6, PC_3_ (γ-GluCys)_3_-Gly); 7, hPC_3_ (γ-GluCys)_3_-Ala); and 8, PC_4_ (γ-GluCys)_4_-Gly). I.S.: internal standard of N-acetyl cysteine (25 nmol per injection).

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
