# Peer review of "The Early Oxidative Stress Induced by Mercury and Cadmium Is Modulated by Ethylene in Medicago sativa Seedlings"

_antioxidants, 2023, doi:10.3390/antiox12030551_

Round 1

Reviewer 1 Report

The manuscript is well-written, the subject is interesting and important. One small point I might make concerns the representation of data in some plots:

- Fig. 1 A-D: the form of line plot is not the best choice I think. It suggest the increase of some value in time. The bar chart would be a better choice. In the same plots: there is no need to report control sample results, while in this case both metal concentration and growth inhibition are obviously equal to zero.

- Fig. 3A and 6A - a multitude of the line makes it difficult to analyze this graph. I would suggest either to divide these plots into two cases - one for Cd results and one for Hg - or to apply different styles of lines for both metals (e.g. every serie for Cd as a solid line and every serie for Hg - dashed line).

With these cosmetic changes made, the manuscript is definitely fit for journal publication.

Author Response

The manuscript is well-written, the subject is interesting and important. One small point I might make concerns the representation of data in some plots:

  1. 1 A-D: the form of line plot is not the best choice I think. I suggest the increase of some value in time. The bar chart would be a better choice. In the same plots: there is no need to report control sample results, while in this case both metal concentration and growth inhibition are obviously equal to zero.

Response: Thanks for this criticism. It is true that the former version of the graph was not clear enough, and we changed it as a bar chart as suggested. We have maintained the control values, which inhibition was negligible (but not completely zero), because of representing mean data, in the growth inhibition graphs (Figs. 1A and 1B), and by means of statistic comparison, to highlight that some treatments after 3 h exposure values were similar to control seedlings. To increase clarity of graphs, we also split some figures and added missing info like molecular weight of relevant proteins in WB immunoblots (new Fig. 2 and Fig. 7).

  1. 3A and 6A - a multitude of the line makes it difficult to analyze this graph. I would suggest either to divide these plots into two cases - one for Cd results and one for Hg - or to apply different styles of lines for both metals (e.g. every serie for Cd as a solid line and every series for Hg - dashed line).

Response: Following the comments of the reviewer, we modified the lines and symbols to allow much better identification of treatments of H2O2 detection by AmplexRed. You can see these changes in the new Figs. 4A and 8A.

Reviewer 2 Report

The manuscript deals with the current issue of the influence of pollutants, specifically hazardous metals, on juvenile alfalfa plants. The obtained results are interesting and stimulating, as pilot results for understanding the relationships and the influence of risk substances on enzymatic activity. I recommend testing the reaction of the plants in the substrate in a subsequent experiment. It is to the detriment of the work that the authors cite older literature rather than newer ones. Are all legacy resources necessary? In the text of the methodology, it is stated that this is Medicago sativa var. Aragon. The correct entry based on the botanical nomenclature is Medicago sativa 'Aragon'. I recommend enlarging Figure 6A. The discussion is rather descriptive. I recommend focusing on analyzing the results.

Author Response

  1. I recommend testing the reaction of the plants in the substrate in a subsequent experiment.

Response: Our experimental approach cannot be extrapolated to real conditions and were performed in juvenile alfalfa plants as was fairly pointed out by the reviewer. Some of the techniques used in our experiment cannot be used directly in mature plants grown in inert substrate (which would allow to treat them at least without edaphic interferences). Our data imply that ethylene signalling is an important feature of metal short-term responses, probably contributing to the onset of oxidative stress symptoms. The mechanism involved in the amelioration of oxidative damage by blocking ethylene perception will be the matter of future research with mature plants, pilot experiments prior its potential use in the phytoremediation of metals under real conditions. Some of these aspects were included in the Discussion section. Please, see below in the response to your question No. 7,

  1. It is to the detriment of the work that the authors cite older literature rather than newer ones. Are all legacy resources necessary?

Response: We added original bibliographic resources to credit their contribution in the field. We have included updated info and relevant articles published recently in the introduction and discussion of our results.

  1. In the text of the methodology, it is stated that this is Medicago sativa Aragon. The correct entry based on the botanical nomenclature is Medicago sativa 'Aragon'.

Response: We have corrected this detail as suggested, besides the source of commercial seeds used.

  1. I recommend enlarging Figure 6A.

Response: Thanks for your suggestion. We made new figures and separated some results to improve clarity.

  1. The discussion is rather descriptive. I recommend focusing on analyzing the results.

Response: We tried to explain our results based on concurrent experiments with different approaches in our manuscript, but also by integrating our data with readily published recent literature. We tried to include an elaborate an integrated Discussion without being too speculative as suggested, in the new text included in the final paragraph of this section:

(Line 543) It is becoming apparent that ethylene mediates the early activation by Hg and Cd of NADPH-oxidase and H2O2 production, as well as the differential changes in GR activity, i.e., inhibition by Hg and modest activation by Cd, and the synthesis of GSH and PCs; symptoms that were transiently delayed in alfalfa seedlings incubated with 1-MCP and exposed for 6 h to Hg and Cd. However, the exact mechanism by which ethylene promotes the induction of NADPH-oxidase activity and the alteration of the redox balance remains unknown, which could involve transcriptional regulation as well as post-translational activation, for example via calcium and calcium dependent kinases (Yamauchi et al., 2017). Future experiments should focus on analysing this aspect of the signalling process that occurs in the early moments of toxic metal exposure by tuning the response to ethylene supplied at different doses and times of incubation, and in the presence of 1-MCP to exploring the behaviour of plants at advanced phenological status (i.e., by comparing juvenile versus adult developmental phases). In addition, the attenuation of the metal-induced oxidative burst was less apparent after 24 h treatment, as Keunen et al. (2015) observed that Arabidopsis plants lacking ethylene response (ein2-1 and ein2-5 mutants) suffered inexorably stress at higher Cd doses or longer treatments. These results imply that other mechanisms of toxicity perception, perhaps depending on other stress related phytohormones, such as jasmonate or abscisic acid (Montero-Palmero et al., 2014b), that may overtake the ethylene dependent pathway as plants become poisoned by toxic elements, which should also be the matter of research. With the idea of improving metal tolerance, an interesting alternative to modify the endogenous levels of ethylene could be the inoculation of plants with bacteria producing ACC deaminase, found to promote plant growth under Cd stress (Ravanbakhsh et al., 2019). Therefore, such strategy could be combined with the selection of plants with limited sensitivity to and/or synthesis of ethylene.

Reviewer 3 Report

The manuscript is overall easy to read and there are no fatal experimental flaws preventing the publication of this study. However, some minor details need to be better explained before the article can be accepted:

1- The number of seeds/plants (replicates) used in each experiment should be detailed in M&Ms.

2- Why was a one-way ANOVA used? From the description, the authors have different factors, so this needs to be explained better. In addition, in my experience, this type of results do not follow a normal distribution. They are usually normalized first. How was this done? 

3- I don't think the word expression is the best one to describe some results. This usually implies the expression of genes or transcripts, which the authors did not use.

4- Ethylene might indeed be involved in stress responses to diminish ROS in Medicago as has been suggested in other species. However, the link between this suggestion and the results is sometimes hard to follow. I think a more obvious solution would have been to test different concentrations of ethylene and check if it indeed the levels of oxidative stress were reduced.

5- A crosstalk between ethylene and nutrients has also been found in other plants. So, a better description of nutrients in M&Ms is at least needed, although the discussion could also benefit from some lines about this. My suggestion here is that ethylene does not work alone to alleviate ROS (as it is implied) but rather in combination with other factors.

Author Response

  1. The number of seeds/plants (replicates) used in each experiment should be detailed in M&Ms.

Response: This information has been clarified in the revised version of the manuscript. Due to the nature of the experiment, we analyse a large number of seedlings per independent biological replicates (25 individuals that were then pooled to constitute a biological independent replicate). You can find more details about the growth conditions used in Supplementary Methods information (supplementary material).

  1. Why was a one-way ANOVA used? From the description, the authors have different factors, so this needs to be explained better. In addition, in my experience, this type of results do not follow a normal distribution. They are usually normalized first. How was this done?

Response: One of the hypotheses tested by a two-way ANOVA analysis of variance is the interaction of two independent variables, normally occurring in a population of samples subjected, for example, to one treatment or condition. In our case, our experimental design ALREADY CONSIDERS THIS INTERACTION, by combining two different conditions (both independent variables): metal toxicity and the incubation of seedlings with the ethylene blocker MCP (four possible combinations or treatments per metal). In consequence, we obtained groups of significantly different means by one-way ANOVA analysis with post-hoc Tukey’s test.

We checked the normal distribution of replicates after one-way ANOVA analysis, that provides different parameters like Shapiro-Wilk test using SPSS software (descriptive statistics). Due to the large number of measurements of different seedlings, we could see that in all cases replicates distributed normally or approximately normally (p < 0.05).

  1. I don't think the word expression is the best one to describe some results. This usually implies the expression of genes or transcripts, which the authors did not use.

Response: We changed this term by suitable terms and checked the were used appropriately when referred to gene transcription along the text.

Following the comments of the reviewer, we modified the lines and symbols to allow much better

  1. Ethylene might indeed be involved in stress responses to diminish ROS in Medicago as has been suggested in other species. However, the link between this suggestion and the results is sometimes hard to follow. I think a more obvious solution would have been to test different concentrations of ethylene and check if it indeed the levels of oxidative stress were reduced.

Response: We did not alter the level of ethylene in our experiments: 1-MCP is an inhibitor of ethylene signalling. Being said this, the reviewer is right: We do not know for sure what cellular mechanism is behind the activation of ROS production triggered by ethylene. It is speculated that NADPH-oxidases could be post-translationally activated. This matches with our data of in gel NADPH-oxidase activity and apoplastic H2O2 release. This was commented in the last paragraph of the Discussion section as explained before in response to Reviewer No. 2.

  1. A crosstalk between ethylene and nutrients has also been found in other plants. So, a better description of nutrients in M&Ms is at least needed, although the discussion could also benefit from some lines about this. My suggestion here is that ethylene does not work alone to alleviate ROS (as it is implied) but rather in combination with other factors.

Response: The reviewer is right: the levels of nutrients may impose changes in the redox and phytohormone balances. It is known that nutrient malnourishment, for example in plants subjected to nitrogen or phosphorous starvation, leads to oxidative stress in roots, possibly through induction of NADPH-oxidases and the contribution of stress-related hormones like ABA and jasmonic acid. However, these interactions cannot be studied in the current experimental design, and must be answered in future research, where we can combine different nutritional status with functional analysis of phytohormone regulation.

Finally, our take-home-message is that ethylene does not work alone, and that other endogenous factors, such as ABA and jasmonic acid, must be considered and studied in future experiments. Please, read the above mentioned last paragraph were we precisely state what the reviewer comments.
